# pH Nonlinearly Dominates Soil Bacterial Community Assembly along an Altitudinal Gradient in Oak-Dominant Forests

**DOI:** 10.3390/microorganisms12091877

**Published:** 2024-09-11

**Authors:** Litao Lin, Guixiang Li, Huiyi Yu, Keming Ma

**Affiliations:** 1Chinese Research Academy of Environmental Sciences, Beijing 100012, China; lin.litao@craes.org.cn; 2Weifang Academy of Agricultural Sciences, Weifang 261071, China; guixiangli2010@163.com; 3State Key Laboratory of Urban and Regional Ecology, Research Center for Eco-Environmental Sciences, Chinese Academy of Sciences, Beijing 100085, China

**Keywords:** altitude, bacteria, environmental filters, law of tolerance, oak forest

## Abstract

Soil bacteria, the predominant microbiota in soil, are subject to the law of minimum and the law of tolerance, but the assembly patterns of soil bacteria in response to environmental factors remain far from clear. Here, we took advantage of an altitudinal gradient (1020–1770 asl) in oak-dominant forests and assessed whether soil bacteria linearly or nonlinearly respond to environmental properties through the changes in the community diversity and composition. We found that soil bacteria decreased with increasing altitude in terms of the species richness and phylogenetic structure, while they were unchanged with increasing altitude in terms of community composition. The species richness was nonlinearly affected by the soil pH (19.9%), C:N ratio (14.3%), SOC (11.4%), and silt + clay content (9.9%). Specifically, the species richness peaked at a pH of 5.5–6.5, and an SOC of 25–50 g kg^−1^, and it showed abrupt decreases and increases at a C:N ratio of 14.5 and a silt + clay content of 70%. The community composition was significantly affected by the soil pH (28.2%), then by the SOC (3.6%), available phosphorus (1.0%), and silt + clay content (0.5%), and it showed less turnovers at a pH of 6.0, SOC of 50 g kg^−1^, and available phosphorus > 3.0 g kg^−1^. These findings imply that environmental filtering processes nonlinearly shape bacterial communities.

## 1. Introduction

Soil bacteria are the most abundant and diverse soil biota, playing critical roles in biogeochemical cycles and ecosystem processes and responding rapidly to environmental changes [1]. The biodiversity pattern of soil bacteria has been widely investigated in recent decades [2,3,4]. However, few studies examined the ecological processes that regulate the soil bacterial community assembly along altitudinal gradients. The distribution of microorganisms is niche-based and exhibits ecological amplitudes in response to changing environments [5,6]. The diversity-stability hypothesis proposes that sudden shifts in biodiversity serve as an important warning sign concerning ecosystem stability. Consequently, elucidating the phylogenetic and taxonomic responses of soil bacteria to environmental factors along an altitudinal gradient is crucial for understanding the potential changes in bacterial communities under global change scenarios. 

Along altitudes, plant communities and soil properties exhibit significant alternations over short distances, which play critical roles in community assembly [7,8]. The plant diversity hypothesis emphasizes that plant production and plant litter heterogeneity affect bacterial diversity and composition [9,10]. The litter quality due to different plant and vegetation types has been demonstrated to significantly affect the composition and activity of bacterial communities [9]. Evidences from arid montane ecosystems indicate that plant diversity is the primary factor affecting the composition of bacterial communities [11]. The niche-based filtering hypothesis posits that soil habitat incompatibility (e.g., pH) causes a decline in the number of bacterial taxa [2,12]. Studies conducted on the Tibetan Plateau confirm that the soil pH plays a pivotal role in promoting the diversity of bacteria and is the primary determinant of bacterial community assembly [4,13]. Neural theory suggests that distance is a significant factor in regulating community assembly [14]. Moreover, some studies also indicate that microbial communities display no diversity difference and evident distance-decay patterns across geological spaces [8,15,16]. Zhang [17], Merino-Martin [18], and Zi [19] suggested that competition in terms of the phylogenetic structure differs among altitudinal gradients. Thus, further comprehensive studies are required to picture the patterns of bacterial community and to identify the environmental factors shaping the process along the altitudinal gradients. Previously, many studies used data from across forest types or vegetation types [4,9], yet the relative importance of these factors in shaping the assembly of bacterial communities within oak forests remains barely studied.

Changes in bacterial diversity and composition may exhibit nonlinear responses to alternations in plant and soil properties. Bacteria typically exhibit ecological amplitudes to changing environment and their assembly could be background-dependent [5,6]. Two of the most significant factors influencing bacterial community are the aboveground plant and soil pH, but little is known to impact the nonlinear responses of a bacterial community to these affecting factors. The majority of previous studies suggested that soil factors (i.e., pH) linearly positively promote or negatively decrease the diversity and/or composition of bacterial communities. With an increasing pH, the growth of bacteria exhibits a “bell-shaped” pattern [6], indicating the nonlinear responses of the bacterial community. Evidence in terms of salinization and altitude confirms that some microbial taxa are observed in a median salinity content along a salinity gradient [12], and dominant bacterial genera have exponentially -increased and exponential-decreased distributions with increasing altitude [3]. In degraded vegetation, the extinction risk of the bacterial community significantly differs compared with the control [20]. Le Roux [21] reported that major bacterial clades exhibit diverse responses to three exotic legume species at invaded sites compared with uninvaded sites. These studies collectively supported the notion that bacterial communities may exhibit ecological amplitudes to changing environments. Accordingly, as an indicator of ecosystem stability, elucidating the nonlinear responses and abrupt changes in the bacterial community to alternations in plant and soil properties could facilitate a more profound insight into the principles of biodiversity conservation.

To address this knowledge gap, soil bacteria from 113 10 m × 10 m plots along altitudinal gradients (from 1020 to 1770 asl) on Dongling Mountain were sampled to determine which factor affects the soil bacterial community assembly: distance, plant, or soil; and the pattern of bacterial communities responding to alternations in plant and soil properties: linear or nonlinear. We hypothesized (1) that bacterial diversity increased and environmental filtering decreased with increasing altitude; (2) that plant diversity rather than soil property was the primary driver of the bacterial community in terms of the diversity and composition; and (3) that the bacterial community exhibited ecological amplitudes to alternations in soil properties in terms of the diversity and composition.

## 2. Materials and Methods

### 2.1. Study Area and Soil Sampling

The experiment site is located in the Beijing Forest Ecosystem Research Station (30°57′29 N, 115°25′33 E), Dongling Mountain, approximately 100 km northwest of Beijing, China. The area is a warm temperate climate zone, with average annual precipitations of 500~650 mm and average annual temperatures of 5~10 °C. The forest in the area is a secondary forest, which is around 80 years old, and is dominated by oak trees (*Quercus wutaishansea*) with a few birches (*Betula* spp.), maples (*Acer mono.*), and shrubs (e.g., *Prunus* spp., *Vitex negundo* var. *hetertophylla*). The studied oak-dominated forest spans an altitudinal gradient of 1020–1770 m, with the treeline occurring at about 1770 m.

Along the western slopes of 10 mountains, 113 10 m width × 10 m height plots with trees were set in 2013 and the 113 plots constituted a continuous altitudinal gradient (i.e., 1020–1770 m asl). In August, the species and abundance of trees in each plot were recorded. Soil samples from besides the tree trunks were then collected at a depth of 0–10 cm. A total of two soil cores (5 cm diameter) (under different tree species) were combined as a single bulk soil sample in each 10 × 10 m plot. The bulk soil samples were sieved through a 2 mm sieve and divided into two sub-samples. One sub-sample was air-dried for the physical and chemical analysis, and the other was kept at −80 °C for DNA extraction.

### 2.2. Soil Characteristics Analysis

The mean soil temperature in August (ST) was recorded using a button thermometer (iButton, 1922L, Maxim Integrated, San Jose, CA, USA) (automatically recording data per hour). Soil moisture (SM) was measured gravimetrically. The soil pH was determined at a ratio of 1:2.5 (soil to water, *w*/*v*). The soil organic carbon (SOC) was determined by the K_2_Cr_2_O_7_ oxidation method [22]. The soil total nitrogen (STN) was measured with a C/N analyzer (Vario EL III, Langenselbold, Germany), and the soil C:N ratio was calculated based on the soil organic carbon and STN. The available phosphorus (AvaP) was measured using the Mo-Sb anti-spectrophotometry method after extraction [23]. The available nitrogen (AvaN) was measured by the alkaline hydrolysis diffusion method [24].

### 2.3. DNA Extraction and Sequencing

The soil DNA was extracted from 0.25 g of freeze-dried soil using a MOBIO Power Soil DNA Extraction Kit (MO Bio Laboratories, Carlsbad, CA, USA) according to the manufacturer’s instructions. The DNA quality was assessed using 260 nm/280 nm and 260 nm/230 nm ratios, and the final DNA concentration was quantified using NanoDrop (Thermo Fisher Scientific, Wilmington, NC, USA). All the extracted DNA was stored at −80 °C prior to utilization.

The V4-V5 region is considered an appropriate choice for accurately reconstructing bacterial phylogeny and avoiding overestimation [25]. The V4–V5 region of the 16s rRNA genes was amplified using the primer pairs 515F/907R (forward primer, 515F, 5′-GTGCCAGCMGCCGCGGTAA-3′; reverse primer, 907R, 5′-GGACTACHVGGGTWTCTAAT-3′), which combined with the barcode sequences. PCR was conducted at a total volume of 25 μL, containing 4 μL 5× FastPfu Buffer, 2 μL 2.5 mM dNTPs, 0.4 μL of each primer (5 mmol·L^−1^), 0.4 μL FastPfu Polymerase (TransGen, Beijing, China), 10 ng template DNA, and dd H_2_O to a final reaction volume of 25 μL. The following cycling parameters were employed: 95 °C for 2 min; 30 cycles at 95 °C for 30 s, 55 °C for 30 s, 72 °C for 45 s; followed by 72 °C for 10 min. The PCR reactions were performed in triplicate for each sample and then pooled together to minimize the potential biases from the amplification. The PCR products were purified using the AxyPrepDNA Gel Extraction Kit (Axygen, Union City, CA, USA). Equal concentrations of amplicons were paired-end sequenced on the Illumina MiSeq PE300 platform (Illumina, San Diego, CA, USA).

### 2.4. Sequence Analysis

The sequence reads were processed with Trimmomatic (Version 0.36) [26], FLASH (Version 1.2.7) [27], and QIIME (Version 1.9.1) software [28]. First, the sequence reads were quality-trimmed using Trimmomatic and merged by FLASH according to the overlap. Using the QIIME software, sequences were removed if they were shorter than 200 bp, ambiguous bases > 0, barcode mismatches > 0, or quality scores < 30 in a window of 50 nt. After quality filtering, the chimeras were checked against the “RDP Gold” database using UCHIME [29] and clustered into operational taxonomic units (OTUs) at a 97% sequence similarity cutoff using the USEARCH (Version 10.0.240) algorithm [29]. The representative OTU sequences were chosen with the most abundant sequences, and then the OTU sequences were aligned to the 13_8 Greengenes database [30] using PyNAST (Version 1.0) [31]. The OTU sequences that were not aligned, non-bacteria, and rare OTUs (sequences < 5) were excluded from the OTUs table and the representative sequences. A phylogenetic tree was constructed in accordance with the standard procedures within QIIME via the use of FastTree. Across the 113 soil samples, 2,263,114 high-quality sequences were obtained, with a range from 9945 to 35,274 in each sample. After a standard dilution at a depth of 9945, 10,813 bacterial OTUs (39 phyla) were identified. Representative sequences have been submitted to the GenBank databases with accession numbers KU147498–KU156629.

### 2.5. Statistical Analysis

The diversity of bacteria is calculated using two metrics, i.e., the phylotype richness (OTUs) and Faith’s phylogenetic diversity [32]. The phylogenetic structure of bacterial communities was assessed using the relatedness index (NRI) [33,34] in the picante package [35]. The NRI is standardized by the mean pair-wise phylogenetic distance (MPD) of all the species pairs in a community and randomized matrices: NRI = –1 × (MPD_obs_ − MPD_rand_)/SD_rand_. The null models by shuffling the tip labels (999 random permutations) among all the species were constructed, and then, Student’s *t*-test (a = 0.05) was performed to determine whether the observed distances between the taxa were greater or less than expected by the null model. Assuming phylogenetic niche conservatism [36], the clustered or overdispersed pattern of the community was determined by assessing whether or not the NRI deviated significantly from the null models [37].

To evaluate the changes in the bacterial OTUs, phylogenetic diversity and NRI according to increasing altitude, we used general linear ordinary least squares regression. To determine how the bacterial OTUs, phylogenetic diversity, and NRI respond to environmental variables, we used random forest analysis in the randomForest package. To determine the changes in the bacterial community composition due to the plant community dissimilarity, environmental distance, and geographic distance, we used partial Mantel test in the ecodist package [38]. Specifically, we measured the relative impact of environmental factors on the community composition by using the partial Mantel test. The pattern of changes in the community composition due to environmental factors was predicted by using generalized dissimilarity modeling (GDM) in the gdm package. All the factors with correlation coefficient *r* ≥ 0.70 are excluded in the models. The Bray–Curtis dissimilarity of the bacterial and plant communities were generated based on the Hellinger-transformed abundance data, as well as the Euclidean distances of the environmental variables (standardized) and geographic distance. All the models were performed with 9999 permutations and analyzed in R statistical software (Version 4.3.1).

## 3. Results

### 3.1. Characterization of Soil Factors, Plant, and Soil Bacteria

The soil temperature significantly decreased with increasing altitude (*p* < 0.05) (Appendix A). The soil moisture showed a significantly convex pattern with increasing altitude (*p* < 0.05). The soil C:N ratio significantly increased with increasing altitude (*p* < 0.05). The soil organic carbon (SOC), total phosphorus (STP), available nitrogen, available phosphorus, and the silt and clay content (silt & clay) showed significantly concave patterns with increasing altitude (*p* < 0.05). Soil pH, total nitrogen (STN), C:P ratio, and electrical conductivity did not show significant patterns along the altitudinal gradient (*p* > 0.05) (Appendix A). SOC exhibited a high Spearman’s correlation with STN (*r* = 0.84, *p* < 0.01), STP (*r* = 0.70, *p* < 0.01), C:P ratio (*r* = 0.71, *p* < 0.01), and available nitrogen (*r* = 0.83, *p* < 0.01) (Appendix A).

The species richness of the trees significantly decreased with increasing altitude (*p* < 0.05) (Appendix A). The abundance of the trees showed a significantly hollow pattern (*p* < 0.05) and the crown density significantly increased with increasing altitude (Appendix A).

Along the altitudinal gradient, 10,813 bacterial OTUs (39 phyla) were identified. The predominant phyla were Proteobacteria, Actinobacteria, Acidobacteria, Planctomycetes, Bacteroidetes, Chloroflexi, and Gemmatimonadetes phyla, accounting for 34.4%, 19.1%, 18.5%, 7.9%, 7.4%, 4.2%, and 3.0% of the bacterial abundance, respectively (Figure 1).

### 3.2. Changes in Bacterial Diversity Associated with Altitude

The species richness and NRI of the bacteria significantly decreased with increasing altitude (*p* < 0.05), while the phylogenetic diversity of the bacteria was not correlated with increasing altitude (*p* > 0.05) (Figure 2A). The majority of bacterial communities in this study were phylogenetically clustered, with NRI values significantly higher than 2 (Figure 2A). The species richness of the bacteria was mainly affected by the soil pH, C:N ratio and litter thickness (*p* < 0.05) (Figure 2B). The phylogenetic diversity of bacteria was mainly influenced by the soil pH, moisture, C:N ratio, and litter thickness (*p* < 0.05).

The changes in the species richness and phylogenetic diversity were not linearly correlated with these soil factors (Figure 3). Specifically, the species richness of the bacteria exhibited higher values at pH of 5.5–6.5 and SOC of 25–50 g kg^−1^, steeper decreases at a C:N ratio of 14.5, and steeper increases at a litter thickness of 3 cm (Figure 3). The phylogenetic diversity of the bacteria showed a similar pattern to that of the species richness, but steeper increases at a soil moisture of 40–45% (Appendix A). The bacterial communities were the least phylogenetically clustered at a pH of 6.0 and C:N ratio of 14.5 and became less phylogenetically clustered with increasing soil moisture and litter thickness (Appendix A).

### 3.3. Changes in Community Composition with Altitude and the Main Impacting Factors

The changes in the bacterial community composition exhibited a significantly positive relationship with the elevational and environmental distances (*p* < 0.05) (Figure 4A,B). When factors were controlled, environmental distance, rather than the elevational distance and tree community dissimilarity showed a significantly partial Mantel relationship with the bacterial community dissimilarity (*p* < 0.05) (Figure 4D). The data from adjacent plots also indicated that the soil pH, rather than the altitude, was the significant factor affecting bacterial community composition (*p* < 0.05) (Figure 5).

The soil pH, SOC, AvaP, and clay were the significant correspondent factors, accounting for 28.2%, 3.6%, 1.0%, and 0.5% of the community composition alternations, respectively (Figure 4E,F). As the GDM results showed, the soil pH, SOC, avaP, and clay showed non-linearly associations with the beta diversity of the bacteria (Figure 6). The beta diversity of the bacteria changed less at a pH of 6.0 and an SOC of 50 g kg^−1^ and changed more at an avaP of 1–3 mg kg^−1^ and a clay + silt of 68–75% (Figure 6).

## 4. Discussion

### 4.1. Bacterial Diversity and Phylogenetic Structure Decreased with Increasing Altitude

This study showed that the bacterial species richness and NRI decreased with increasing altitude. This result partially supported hypothesis 1, namely the bacterial diversity and environmental filtering increased with increasing altitude. Decreases in the bacterial species richness with increasing altitude were also observed in the Smoky Mountains [39], lake sediments [40], and southeastern Tibetan Plateau [3]. The result differed from our previous work in herbaceous soils where the bacterial diversity had a “hollow-shaped” distribution with increasing altitude [2]. The differences could be caused by the “hollow-shaped” pattern of herbaceous diversity. As the dominant driving factors (e.g., pH) showed no significant relationship with increasing altitude, the linearly decreasing pattern could be attributed to the decrease in tree richness and litter thickness (Appendix A). The plant diversity hypothesis supports the notion that carbon resource availability and heterogeneity have large filtering effects on bacterial diversity [9,10], which is consistent with the decreased pattern of bacterial diversity in this study. The majority of soil bacterial communities in this study were phylogenetically more closely related than expected by chance (Figure 2). Similar to our results, previous studies have revealed clustered phylogenetic structure patterns of bacteria sampled from soil, marine sediments or freshwater [41] at global or regional scales. The heterogeneity in litter properties and root traits due to high tree diversity contributed to biotic filtering in the bacterial community [18], thus, the biotic filtering became weaker towards high altitude. Increases in the C:N ratio from 10:1 to 14.5:1, which is the main C:N ratio range of soil samples along the altitude, could reduce the restriction of carbon resources to bacteria [42], thus decreasing the NRI toward high altitude. Zhang [17] also indicated that the NRI of bacteria in Tibet is significantly explained by the SOC. Thus, the strength of clustering and perhaps environmental filtering may decrease with increasing carbon and nitrogen availability along altitude. Similar to this study, Shen [4] suggested a negative correlation between the strength of environmental filtering and altitude in alkaline soils. In contrast to the positive altitudinal pattern of environmental filtering found by Zeng [40] and Zi [19], the NRI in this study significantly decreased with increasing altitude (Figure 2) and litter thickness (Appendix A) and increased with tree richness (Appendix A), indicating that the filtering of tree properties may contribute to the decreased pattern of the soil bacterial phylogenetic structure.

### 4.2. Bacterial Community Restricted by Soil Condition over Plant Property

This study showed that the assembly of bacterial communities was more regulated by the soil properties over the plant communities in terms of the diversity and composition. The soil properties (i.e., pH, SOC, and available P), rather than the spatial distance or tree community dissimilarity, demonstrated significantly partial Mantel relationships with the bacterial community composition (Figure 3). The results did not support hypothesis 2, namely that plant diversity was the primary driver of the bacterial community diversity and composition. Trees can alter the soil environment through litter fall and biotic interaction, thereby exerting an indirect influence on the bulk soil bacteria [43]. In contrast to rhizosphere bacteria that are strongly associated with the root trait diversity and density [18], it may be difficult to discern a direct relationship between trees and the bulk bacterial community after controlling for the soil properties (Figure 4). Furthermore, the composition of the bacterial community did not exhibit a distance-decay pattern after controlling for the soil properties (Figure 3), which contrasts with the findings of the previous study on soil *Acidobacteria* [16]. The dispersal ability of a soil bacteria appears to be unconstrained by distance at an altitudinal gradient [16], suggesting that the clustered phylogenetic structure observed may be caused by abiotic environment rather than a dispersal limitation.

The soil pH is identified as the primary factor shaping the bacterial diversity and community composition, accounted for 20% and 18% of the variations, respectively (Figure 2 and Figure 3). The soil pH imposes physiological constraints on the assembly of bacterial communities [2,44]. Previous studies have reported that the pH preference trait of bacteria is relatively deeply conserved [45], and thus high pH environments may filter towards more closely related bacterial species in the oak-dominant forest. At a pH of 6.0, the bacterial community showed relatively higher species richness values and lower NRI values (Figure 3 and Appendix A), indicating that the phylogenetic structure was associated with the soil pH. A previous study in this area reported that the pH negatively correlated with the phylogenetic diversity of the soil bacteria in the herb layer, indicating evident impacts of the pH on the bacterial phylotypes along pH gradient [2]. Similarly, the pH filtering patterns were also observed in a soil diazotrophic community [13]. Cho [46] also reported that soil pH, rather than the elevation, was the determining factor in the assembly of the bacterial phylogenetic communities on Mount Norikura.

Apart from the soil pH, the soil nutrient condition may serve as a significant restricting factor in the phylogenetic structure of bacteria. The SOC and available P were highly correlated (Appendix A) and accounted for 3.6% and 1.0% of the community composition, respectively (Figure 4). The habitat hypothesis states that the number of non-dominant microbial species declines as the habitat incompatibility increases [47]. Increases in the SOC could improve the habitat quality of soil bacteria and promote the habitat heterogeneity [17]. Thus, more restrictive SOC and available P resources would facilitate the size and diversity of microbial communities, particularly for non-dominant groups. Similar to our result, Lin et al. (2024) reported that bacterial communities were promoted by the SOC in terms of the diversity and composition [48]. Overall, these results demonstrate that soil acidity, organic carbon, and nutrient resources are the main abiotic filters structuring bacterial communities along the altitudinal gradient in the oak-dominated forest.

### 4.3. “Bell-Shaped” Responses of Bacterial Community to Increasing Soil pH and Nutrients

The changes in the bacterial species richness and community composition were background-dependent and nonlinearly correlated with the soil pH. The species richness of the bacterial community firstly increased and then decreased with an increasing pH from 5.2 to 7.69, with relatively higher values at a pH of 5.8–6.5 (Figure 3). This result, from the perspective of ecological amplitudes, deepened our understanding of the responses of the bacterial community to environmental changes and supported the hypothesis 3, namely that bacterial community has ecological amplitudes. The result differed from previous studies that found the bacterial community assembly linearly changed with soil pH in term of species richness and composition [4,13]. This could be attributed to the fact that the oak forests have neutral soils and that the bacterial community in oak forests may have an appropriate pH range for activity. Along extensive environmental gradients, studies have reported that changes in bacterial community composition were hump-shaped [49]. The acidic or alkaline pH soils provided stronger physiological constraints on the survival of the bacterial taxa compared with the neutral pH soils, due to greater energy costs associated with homeostasis [50]. The bacterial taxa in this study showed relatively lower filtrated patterns at neutral pH soils, with lower NRI values at pH~6.0 (Appendix A). The GDM results also showed that bacterial communities had lower turnover rates at median pH compared with those at high or low pH levels (Figure 5). Similar to our result, the activity of the bacterial communities suggests a “bell-shaped” pattern with an increasing pH [6,51]. Along large pH gradients, Mod [52] reported that bacterial communities in soils with a pH of 6.0–7.0 soils showed higher species richness levels than those in acid and alkaline soils. In acid habitats, phylogenetic structure of the bacterial community became less clustered with increasing pH [46] and the bacterial diversity were positively associated with the pH [3,53]. In alkaline habitats, Xiong [54] reported that bacterial community decreased with an increasing pH in terms of diversity and phylogenetic diversity. Therefore, bacterial community could have ecological amplitudes along the pH gradient ranging from 5.2 to 7.69 and show nonlinear variations to an increasing pH.

## 5. Conclusions

This study showed that most bacterial communities along the altitude in gradient the oak-dominant forest were phylogenetically clustered, and that the niche-based filtering was mainly explained by the soil properties (i.e., pH, C:N ratio, and SOC) rather than the elevation difference and plant properties. Soil bacteria exhibited nonlinear variations with increasing soil pH, C:N ratio, and SOC in terms of species richness and community composition. Specifically, the soil bacteria had the highest species richness and lower species turnovers at a pH of 5.5–6.5 and an SOC of 25–50 g kg^−1^, and abrupt decreases in species richness and higher species turnovers at a C:N ratio of 14.5. These results broaden our understanding of the niche-based filtering of bacteria in oak forests and confirm the law of minimum and the law of tolerance in regulating bacterial communities.

## Figures and Tables

**Figure 1 microorganisms-12-01877-f001:**
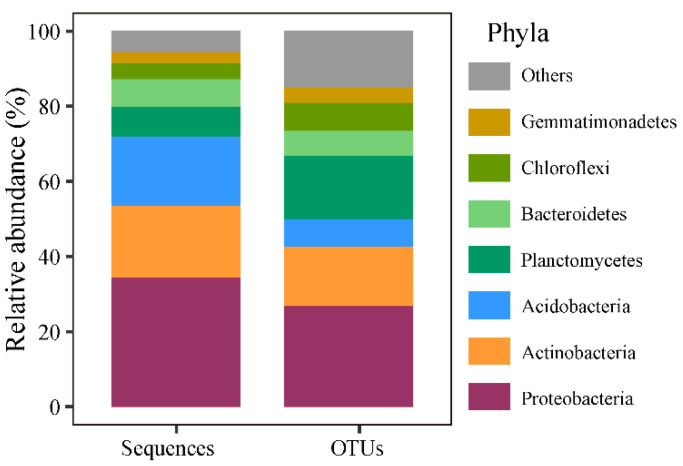
Relative abundance of the sequences and the OTUs of dominant bacteria phyla.

**Figure 2 microorganisms-12-01877-f002:**
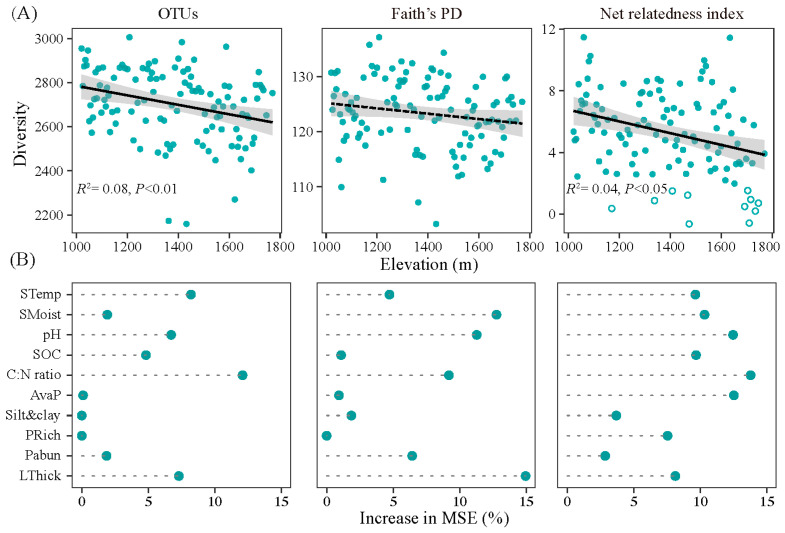
Changes in bacterial diversity associated with altitude. (**A**), Altitudinal pattern; and (**B**) relative importance (% of the mean increase in mean square errors, MSE) of the environmental factors. Positive net relatedness index (NRI) values (solid circles, *p* < 0.05) indicate phylogenetic clustering, and negative NRI values (open circles) indicate phylogenetic overdispersion.

**Figure 3 microorganisms-12-01877-f003:**
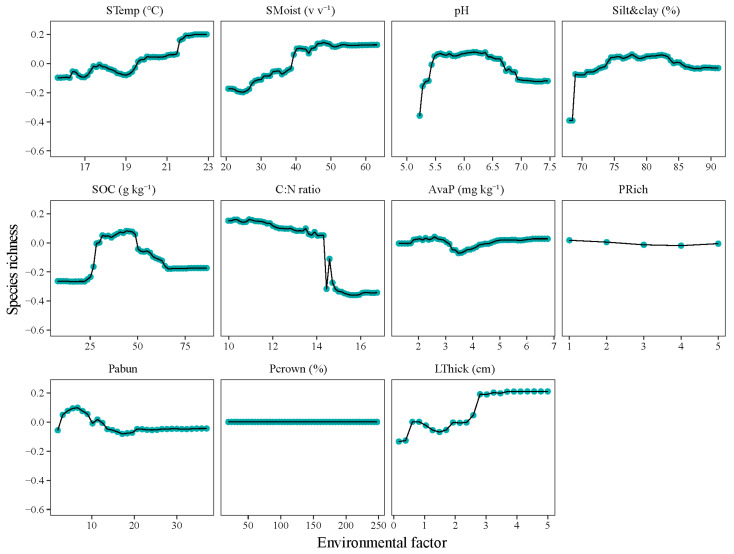
Partial dependence of plant, and soil properties on the species richness of soil bacteria using random forest modeling. STemp, soil temperature; SMoist, soil moisture; SOC, soil organic carbon; C:N ratio, soil organic carbon to total nitrogen ratio; AvaP, soil available phosphorus; PRich, species richness of plant; Pabun, abundance of plant; Pcrown, crown density of plant; LThick, litter thickness.

**Figure 4 microorganisms-12-01877-f004:**
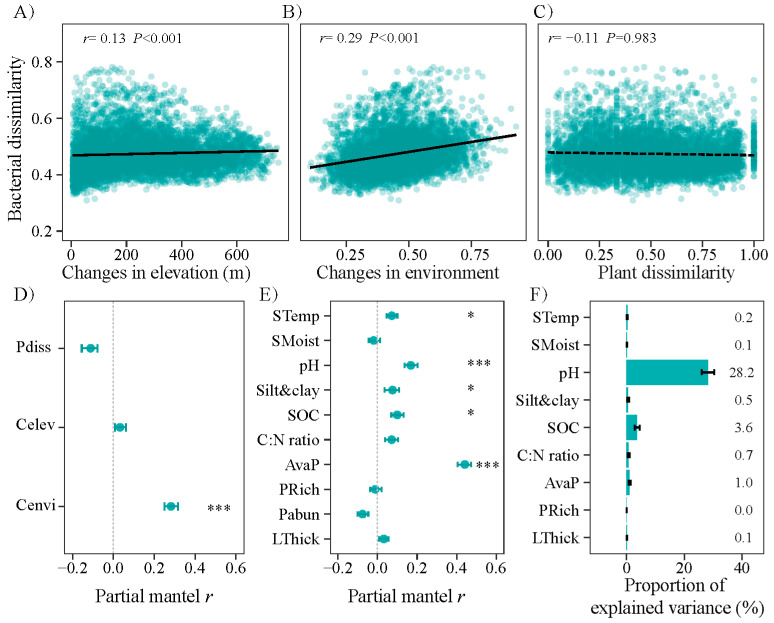
Changes in community composition dissimilarity of soil bacteria according to altitudinal difference, environmental distance, plant dissimilarity, and the affecting factors. (**A**–**C**) Solid and dashed lines denote significant and insignificant mantel correlations. (**D**–**F**) Error bars denote the 95% confidence intervals. *, *p* < 0.05; ***, *p* < 0.001.

**Figure 5 microorganisms-12-01877-f005:**
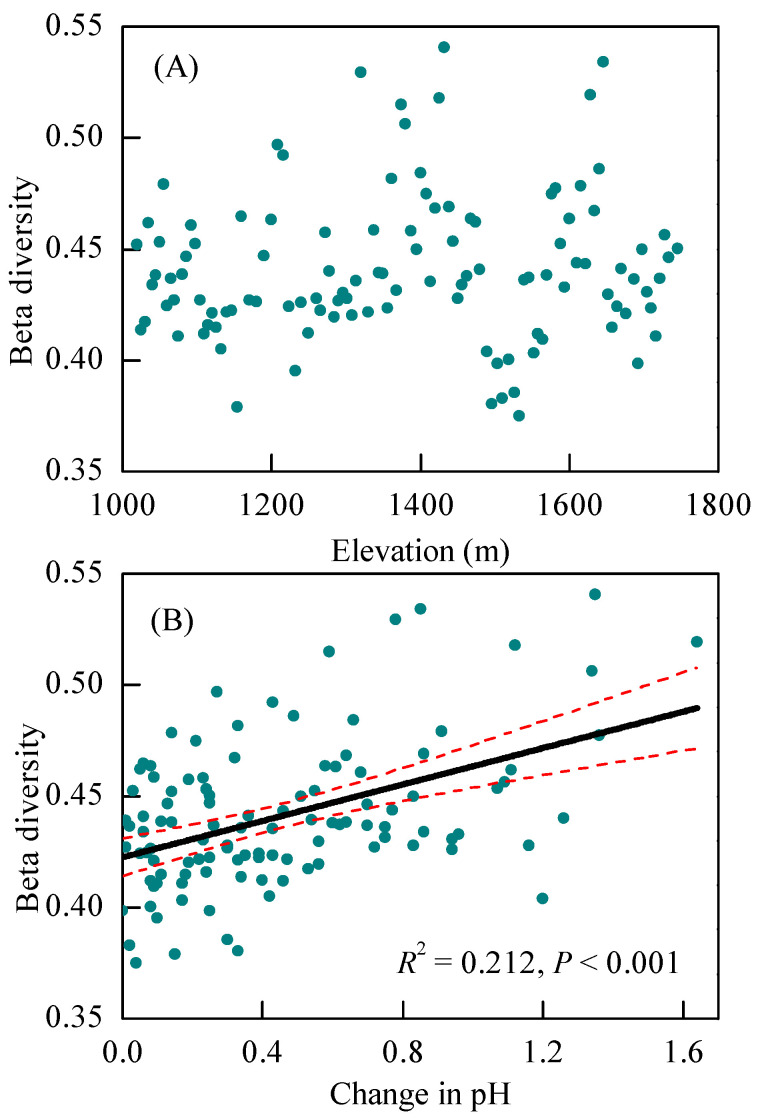
(**A**) Changes in community composition dissimilarity according to increasing altitude, (**B**) soil pH based on adjacent sampling sites. Solid and dashed lines denote the significant regression and 95% confidence intervals, respectively.

**Figure 6 microorganisms-12-01877-f006:**
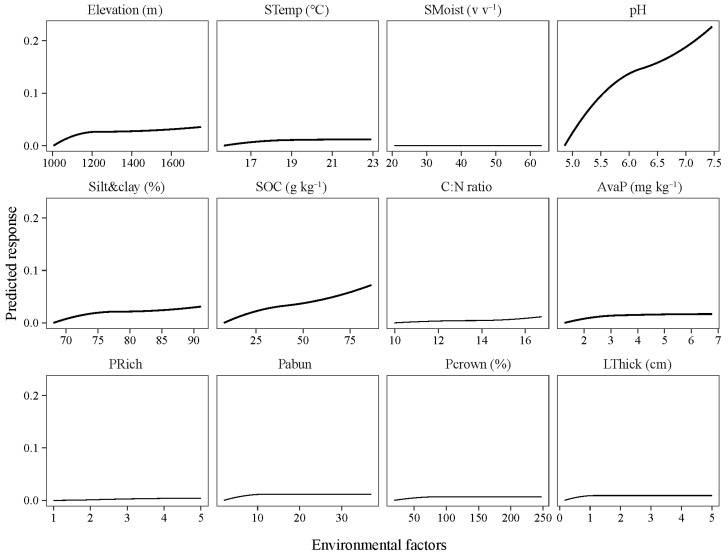
Predicted response of community composition to increasing altitude and environmental factors. STemp, soil temperature; SMoist, soil moisture; SOC, soil organic carbon; C:N ratio, soil organic carbon to total nitrogen ratio; AvaP, soil available phosphorus; PRich, species richness of plant; Pabun, abundance of plant; Pcrown, crown density of plant; LThick, litter thickness.

## Data Availability

Data are contained within the article.

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
