# Peer review of "pH Nonlinearly Dominates Soil Bacterial Community Assembly along an Altitudinal Gradient in Oak-Dominant Forests"

_microorganisms, 2024, doi:10.3390/microorganisms12091877_

Round 1

Reviewer 1 Report

Comments and Suggestions for Authors

Lin and collaborators' manuscript " pH nonlinearly dominates soil bacterial community assembly along an altitudinal gradient in oak-dominant forests" aims to determine the factors driving soil bacterial assembly and the nonlinear patterns through which bacterial communities respond to changes in soil properties.

The manuscript is well-written and well-organized. The methods are robust, and the statistical analyses are sound. 

There are only some minor comments: 

L62 – Check redaction (litter)

L95 – Specify if the number 119 is the total sample size. In L146, it is specified that 113 soil samples were analyzed. Clarify

L104 – Specify if the field (soil temperature) was determined during soil sampling (August 2013). 

L105 – How long were the samples stored till determination? (weeks or years?)

L194 – Replace: Changes in bacterial diversity associated with altitude. Same comment to L221; check grammar. Some connection words are missing; “to” is not enough. 

Author Response

## Reviewer1

Lin and collaborators' manuscript "pH nonlinearly dominates soil bacterial community assembly along an altitudinal gradient in oak-dominant forests" aims to determine the factors driving soil bacterial assembly and the nonlinear patterns through which bacterial communities respond to changes in soil properties.

The manuscript is well-written and well-organized. The methods are robust, and the statistical analyses are sound. 

 Reply: Thank you very much for taking the time to review our manuscript and providing constructive comments and suggestions! We addressed your concerns one by one as follows.

There are only some minor comments: 

  1. L62 – Check redaction (litter)

Reply: Thanks for the suggestion. We had a thorough review of the editing of the manuscript. E.g., in 2.4 subsection, the “OTUs were picked representative sequences were chosen with the most abundant sequences, and then reference sequences of OTUs were assigned taxonomies using the RDP classifier method against the 13_8 Greegenes database [30]. Sequences were aligned to the Greengenes reference alignment using PyNAST [31], and a phylogenetic tree was constructed with FastTree according to the standard procedures within QIIME” were shorten as “The representative OTU sequences were chosen with the most abundant sequences, and then reference sequences of OTU sequences were assigned taxonomies using the RDP classifier method against the 13_8 Greegenes database [30]. Sequences were aligned to the 13_8 Greengenes database [30] reference alignment using PyNAST [31]”.

  1. L95 – Specify if the number 119 is the total sample size. In L146, it is specified that 113 soil samples were analyzed. Clarify

Reply: A total of 113 plots with trees were set in this study. There are six plots with no tree along the altitudinal gradient. Therefore, the data of 113 plots were used in the calculation in the study.

In the second paragraph of 2.1 Study area and soil sampling, it is clarified as “113 10 m width × 10 m height plots with trees were set in 2013 and the 113 plots constituted a continuous altitudinal gradient”.

In the last paragraph of the introduction, it is clarified as “soil bacteria of 113 10 m * 10 m plots along altitudinal gradients (from 1020 to 1770 asl) in Dongling Mountain were sampled”.

  1. L104 – Specify if the field (soil temperature) was determined during soil sampling (August 2013). 

Reply: The soil temperature were recorded by using a button thermometer which automatically records temperature pen hour. At line 104, the sentence is clarified as “The mean soil temperature in August (ST) was recorded using a button thermometer (iButton, 1922L, Maxim Integrated) (automatically recording data pen hour)”.

  1. L105 – How long were the samples stored till determination? (weeks or years?)

Reply: It is about four months. The soil was sampled in 2013 August and the soil DNA were extracted and amplified before the end of year 2013 and then send for sequencing on Miseq PE300 platform.

  1. L194 – Replace: Changes in bacterial diversity associated with altitude. Same comment to L221; check grammar. Some connection words are missing; “to” is not enough. 

Reply: As suggested, the titles of the 3.2 subsection and figure 2 were replaced with “Changes in bacterial diversity associated with altitude” at the line 194 and line 221. We thoroughly checked the grammar and added missing connection words. For example, in 3.3 subsection, we added “to environmental changes” behind “The result, from perspective of ecological amplitudes, deepened our understanding of the responses of bacterial community”.

## Reviewer2

The manuscript "pH nonlinearly dominates soil bacterial community assembly along an altitudinal gradient in oak-dominant forests" present a well organized field study, with multiple findings and results.

The authors propose an altitudinal gradient as factor, a high number of plots for sampling and triplicates for each plot. Based on the clear hypotheses formulated at the end of the Introduction, the manuscript present a well described and detailed flow of results. 

The entire work of the authors is organized and the findings are completely explored.

Reply: thanks for your positive comments!

Reviewer 2 Report

Comments and Suggestions for Authors

Dear authors,

The manuscript "pH nonlinearly dominates soil bacterial community assembly along an altitudinal gradient in oak-dominant forests" present a well organized field study, with multiple findings and results.

The authors propose an altitudinal gradient as factor, a high number of plots for sampling and triplicates for each plot. Based on the clear hypotheses formulated at the end of the Introduction, the manuscript present a well described and detailed flow of results. 

The entire work of the authors is organized and the findings are completely explored.

Author Response

## Reviewer2

The manuscript "pH nonlinearly dominates soil bacterial community assembly along an altitudinal gradient in oak-dominant forests" present a well organized field study, with multiple findings and results.

The authors propose an altitudinal gradient as factor, a high number of plots for sampling and triplicates for each plot. Based on the clear hypotheses formulated at the end of the Introduction, the manuscript present a well described and detailed flow of results. 

The entire work of the authors is organized and the findings are completely explored.

Reply: thanks for your positive comments!  we  We had a thorough review of the editing and grammer to improve the qualtiy of manuscript.  
